# Effects of Extraction Process Factors on the Composition and Antioxidant Activity of Blackthorn (*Prunus spinosa* L.) Fruit Extracts

**DOI:** 10.3390/antiox12101897

**Published:** 2023-10-23

**Authors:** Ana-Maria Drăghici-Popa, Aurelian Cristian Boscornea, Ana-Maria Brezoiu, Ștefan Theodor Tomas, Oana Cristina Pârvulescu, Raluca Stan

**Affiliations:** 1Department of Organic Chemistry, National University of Science and Technology POLITEHNICA Bucharest, 1-7 Gheorghe Polizu St., 011061 Bucharest, Romania; ana_maria.draghici@upb.ro; 2Department of Bioresources and Polymer Science, National University of Science and Technology POLITEHNICA Bucharest, 1-7 Gheorghe Polizu St., 011061 Bucharest, Romania; cristian.boscornea@upb.ro (A.C.B.); aastomas@hotmail.com (Ș.T.T.); 3Department of Chemical and Biochemical Engineering, National University of Science and Technology POLITEHNICA Bucharest, 1-7 Gheorghe Polizu St., 011061 Bucharest, Romania; anamaria.brezoiu@gmail.com

**Keywords:** *Prunus spinosa* L., antioxidant capacity, phenolic compounds, anthocyanins, regression model, optimization

## Abstract

This study aimed at establishing the optimal conditions for the classic extraction of phenolic compounds from *Prunus spinosa* L. fruits. The effects of different parameters, i.e., ethanol concentration in the extraction solvent (mixture of ethanol and water), operation temperature, and extraction time, on process responses were evaluated. Total phenolic content (*TPC*), total anthocyanin content (*TAC*), antioxidant capacity (*AC*), and contents of protocatechuic acid (*PA*), caffeic acid (*CA*), vanillic acid (*VA*), rutin hydrate (*RH*), and quercetin (*Q*) of fruit extracts were selected as process responses. A synergistic effect of obtaining high values of *TPC*, *TAC*, *AC*, *PA*, and *VA* was achieved for the extraction in 50% ethanol at 60 °C for 30 min. At a higher level of process temperature, the extraction of protocatechuic acid and vanillic acid was enhanced, but the flavonoids, i.e., rutin hydrate and quercetin, were degraded. A lower temperature should be used to obtain a higher amount of flavonoids. *TPC*, *TAC*, *AC*, and phenolic acid contents (*PA*, *CA*, and *VA*) in the extract samples obtained at an ethanol concentration of 50–100%, a temperature of 30–60 °C, and an extraction time of 30 min were strongly directly correlated.

## 1. Introduction

Due to the tendency of consumers to choose foods with a positive impact on health, producers have to develop new foods with enhanced beneficial properties. These actions can be carried out in two ways: the former refers to returning to traditional products, while the latter focuses on the production of functional foods by enriching foods with unconventional additives or constituents/ingredients. One method of obtaining the necessary bioactive compounds for functional foods is their extraction from vegetal sources that are not usually valued, such as wild fruits [1,2]. 

*Prunus spinosa* L., which belongs to the rose family (Rosaceae), grows as a shrub in wild areas of Europe and temperate regions of Asia and Mediterranean countries [1,3,4,5]. *P. spinosa* fruits are known as blackthorn, grater, rasp, sloe, or stickleback and are closely related to plums, cherries, peaches, nectarines, and apricots [6]. They are used in phytotherapy for the treatment of many conditions related to different forms of cough, are mildly laxative, diuretic, spasmolytic, anti-inflammatory, and have an antiseptic effect [3,7]. These fruits can also be used to prepare jams or macerated with anise liqueur to obtain a digestive alcoholic drink called patxarán [7].

*P. spinosa* fruits contain different polyphenolic compounds, such as phenolic acids, flavonoids, anthocyanins, coumarins, and proanthocyanins [7,8,9,10], which determine their antioxidant [9,10,11,12,13], antimicrobial [14,15], anti-inflammatory [4,13], and anticancer [16] properties. Despite all these beneficial properties, it is difficult to find innovative food products on the market. However, a lot of interesting recipes can be found in traditional cuisine and folk medicine [2]. 

Polyphenols are a class of natural compounds that play an important role in plant development and also contribute to the sensory characteristics of fruits and vegetables [17,18]. They are very uncommon to found in free form, most of them being isolated in conjugated forms, most often having linked glycosidic residues. 

Phenolic compounds represent important secondary metabolites of plants characterized by one or more aromatic rings and several attached hydroxyl groups. These compounds provide protection to the plant against pathogens, UV rays, parasites, and free radicals [17,19,20]. They are classified into four main categories, i.e., flavonoids (including flavonols, flavones, isoflavones, flavanones, anthocyanidins, and flavanols), phenolic acids (including compounds derived from hydroxybenzoic acids, such as gallic acid, and those derived from hydroxycinnamic acid, e.g., caffeic, ferulic, and coumaric acids), stilbenes, and lignans [18,19,21]. Phenolic compounds present a wide range of beneficial properties for human health, especially antiallergenic, anti-inflammatory, antimicrobial, antioxidant, cardioprotective, and vasodilator properties [17,20,22].

The antibacterial, antifungal, and anticancer effects of polyphenolic extracts from *P. spinosa* fruits and their possible applications in the development of new pharmaceutical products have been evaluated in several studies. An aqueous extract showed antibacterial effects on a strain of *Pseudomonas* sp. [14], whereas an ethanolic extract had antimicrobial activity against *Staphylococcus aureus*, *Escherichia coli*, *Pseudomonas aeruginosa*, *Salmonella abony*, and *Candida albicans* [3]. Also, a methanolic extract from *P. spinosa* fruits presented a significant antioxidant capacity and led to an important decrease in glioblastoma from brain cancer cells [23]. 

In the last two decades, the interest in investigating these compounds has been shown by numerous papers related to the extraction and separation of phenolic compounds [24]. They are widely considered to be highly unstable and highly susceptible to degradation. The stability of polyphenols under different conditions is a very important aspect that must be taken into account to ensure the desired properties of the final product and maintain the biological activity and structure of the compounds during the different processing stages [25,26]. The main parameters influencing the performance of the extraction process are extraction time, temperature, solvent type/composition, solvent-to-solid ratio, and number of extraction stages. To remove unwanted compounds, e.g., waxes, fats, terpenes, and chlorophylls, additional stages of extraction of these compounds can be introduced [24,27]. Modern extraction techniques include high extraction temperatures to increase extraction yield, but the thermal sensitivity of polyphenols, especially anthocyanins, must be considered. According to the literature, polyphenols do not withstand temperatures higher than 100 °C for more than 1 min without severe loss of their activity [26]. The stability of standard polyphenols and plant extracts against UV irradiation is relatively high. The highest stability was observed for gallic acid and vanillic acid [25]. Anthocyanins are easily oxidized and therefore susceptible to oxidative degradation during various stages of processing and storage. There are several factors that affect the stability of anthocyanins and, implicitly, of products containing anthocyanins, including pH, temperature, light, oxygen, metal ions, enzymes, and sugars [26,28].

Due to its simplicity, high efficiency, and large-scale applicability, solid–liquid extraction is still the most commonly used extraction method [24]. It involves the use of conventional solvents, e.g., alcohols (methanol, ethanol), acetone, diethyl ether, and ethyl acetate, often mixed with various proportions of water. There are several disadvantages to using these solvents. In addition to a possible dangerous effect on human health, solvent residues can also remain in the final products [17,24,29]. Thus, in this paper, mixtures of ethanol, a green solvent, and water were selected to obtain polyphenolic extracts from *P. spinosa* (blackthorn) fruits with enhanced antioxidant properties.

The aim of this work was to establish the optimal conditions for the classic extraction of polyphenols from Romanian blackthorn fruits. The effects of extraction solvent composition, process temperature, and extraction time on the total phenolic content, total anthocyanin content, antioxidant activity, and chemical profile of extracts (contents of protocatechuic acid, caffeic acid, vanillic acid, rutin hydrate, and quercetin) were evaluated.

## 2. Materials and Methods

### 2.1. Plant Material

Blackthorn fruits were harvested in October 2019 from Giurgiu County (Romania). After the stones were removed, the fruits were oven-dried at 30 °C until a constant mass was obtained. The moisture contents of fresh and dried fruits were 76 ± 1% and 7.0 ± 0.2%, respectively. The size reduction of dried fruits was achieved using an electric grinder. Dried and ground fruits were stored in a dry place until used.

### 2.2. Chemicals

Ethanol, isooctane, and lactic acid were of analytical grade and were provided by Merck KGaA (Darmstadt, Germany). Folin–Ciocalteu reagent, gallic acid, and sodium carbonate required for spectrophotometric determination as well as copper (II) chloride, ammonium acetate, neocuproine, and Trolox used for cupric reducing antioxidant capacity (CUPRAC) analysis were purchased from Merck KGaA. 

For chromatographic analyses, several standard substances were used, i.e., caffeic acid (98%, HPLC grade, Merck KGaA), caftaric acid (Molekula GmbH, München, Germany), catechin hydrate (>98%, HPLC grade, Merck KGaA), chlorogenic acid (primary reference standard, HWI Group, Ruelzheim, Germany), chicoric acid (>98%, TCI, Tokyo, Japan), cyanidin chloride (>95%, HPLC grade, Merck KGaA), delphinidin chloride (analytical standard, Merck KGaA), gallic acid (98%, Alfa Aesar, Haverhill, MA, USA), (−) epicatechin (>98%, HPLC grade, TCI), ellagic acid dihydrate (>98%, HPLC grade, TCI), gallic acid (98%, Alfa Aesar), kaempferol (>97%, HPLC grade, Merck KGaA), malvidin chloride (>95%, HPLC grade, Merck KGaA), myricetin (>96%, HPLC grade), pelargonidin chloride (Merck KGaA), protocatechuic acid (>98%, HPLC grade, TCI), quercetin (>95%, HPLC grade), rosmarinic acid (>98%, HPLC grade, Merck KGaA), rutin hydrate (95%, HPLC grade), syringic acid (>98.5%, Molekula GmbH), trans-p-coumaric acid (analytical standard, Merck KGaA), trans-ferulic acid (>98%, GC), and trans-resveratrol (certified reference material, Merck KGaA). 

### 2.3. Extraction Procedure

Firstly, the lipids found in fruits were removed by extraction with isooctane for 2 h at room temperature. The mixture was filtered, and the solid phase was dried prior to its use for the extraction of polyphenolic compounds. Mixtures of ethanol and water acidified with 0.6% (*v*/*v*) lactic acid were used as extraction solvents. Ethanol concentration in the extraction solvent (*c_et_*), extraction temperature (*t*), and extraction time (*τ*) were selected as process independent variables (factors). Extraction experiments were performed at different levels of *c_et_* (50, 66.67, 75, and 100% *v*/*v*), *t* (30, 40, 50, 60, and 82.5 °C), and *τ* (5, 15, 30, 60, 120, and 180 min), using a heating plate equipped with a temperature control unit and a magnetic stirrer. The solvent-to-fruit ratio of 10:1 (*v*/*w*) and a constant stirring rate of 1100 rpm were used for all experiments. After the extraction, the mixtures were centrifuged at 4000 rpm for 10 min, and the supernatants were kept at a temperature of 4 °C before the analysis was performed. 

### 2.4. Total Phenolic Content (TPC)

The *TPC* in the extracts was determined using the Folin–Ciocalteu method. An extract sample (0.25–0.5 mL), Folin–Ciocalteu reagent (1 mL), distilled water (3 mL), and 20% Na_2_CO_3_ solution (1.5 mL) were added to a 10 mL volumetric flask. The mixtures were kept in the dark for 30 min, and then their absorbance was measured at 750 nm using a Jasco V 550 UV-Vis spectrophotometer (Jasco Corporation, Tokyo, Japan). The results were expressed as mg of gallic acid equivalents (GAE) per g of dry matter (mg GAE/g DM) using a standard curve corresponding to 0–1.26 mg gallic acid/mL. The analysis of each extract was performed in triplicate. 

### 2.5. Total Anthocyanin Content (TAC)

To determine the *TAC*, the samples were diluted appropriately with the extraction solvent, and the absorbance was measured at 520 nm using a Jasco V 550 UV-Vis spectrophotometer. *TAC* was expressed as mg of cyanidin-3-glucoside equivalents (C3GE) per g of dry matter (mg C3GE/g DM) and calculated using Equation (1) [30], where *A* is the sample absorbance, *M* the molecular mass of cyanidin-3-glucoside (449.2 g/mol), *F_D_* the dilution factor of the extract, *V* the volume of the extract (L), 1000 the factor for conversion from g to mg, *ε* the molar extinction coefficient of cyanidin-3-glucoside (26,900 L/(mol·cm)), *L* the cuvette length (cm), and *m* the mass of the dry matter (g). The analyses were performed in triplicate.
(1)$$TAC=\frac{1000AMF_{D}V}{\varepsilon Lm}$$

### 2.6. Antioxidant Capacity (AC)

The *AC* was determined using the CUPRAC method [31]. For sample analysis, 1 mL copper (II) chloride, 1 mL ammonium acetate buffer solution, 1 mL neocuproine, *x* mL of sample, and (1.1 − *x*) mL of distilled water were added to a vial so that the total volume was 4.1 mL. The absorbance was measured after 30 min at 450 nm. The analysis was performed in quartz cuvettes using a Jasco V 550 UV-Vis spectrophotometer. The results were expressed as µmol of Trolox equivalents (TE) per g of dry matter (µmol TE/g DM) using a standard curve corresponding to 0–0.25 mg Trolox/mL. The analyses were performed in triplicate.

### 2.7. Chemical Profile of Extracts

High-performance liquid chromatography (HPLC) analyses were performed using a Shimadzu Nexera-2 system (Shimadzu Corporation, Kyoto, Japan) equipped with a photodiode array (PDA) detector (SPD-M30A) with a Nucleosil C18 reversed-phase separation column (2.7 μm × 2.7 μm × 100 mm). The analyses were performed at a flow rate of 0.4 mL/min using gradient elution having as mobile phases H_2_O/formic acid (100/2.5) (*v*/*v*) and ACN/H_2_O/formic acid (90/10/2.5) (*v*/*v*/*v*), using 1 μL as injection volume for every sample. The gradient elution program as well as calibration curves for standard substances are described elsewhere [32,33]. The identification of each component of the extract was performed considering the retention time and the similarity of the UV-Vis spectrum compared to that of standard substances, and the quantification was performed using calibration curves at the maximum absorption wavelength for each standard substance. Data regarding standard substance retention times, quantification wavelength, calibration curves, and limits of detection and quantification are summarized in Appendix A. Before analysis, all extracts were filtered through a 0.45 μm nylon syringe filter and analyzed without further dilution, with three successive injections. Their chemical profile was determined as an average of three replicates and expressed as mg/100 g DM.

High-resolution mass spectrometry (HRMS) analyses were performed using a Fourier-transform ion cyclotron resonance (FT-ICR) spectrometer (SolariX XR 15T, Bruker Daltonics, Bremen, Germany) equipped with an electrospray ionization (ESI) source. Each sample was introduced by direct infusion, using negative ion mode ESI and the following parameter levels: sample flow rate of 120 µL/h, nebulizer gas (N_2_) pressure of 2.8 bar and flow rate of 3 L/min at a temperature of 200 °C. The spectra were recorded over a mass range between 92 and 1500 AMU at a source voltage of 4700 V.

### 2.8. Statistical Analysis

All experimental measurements were performed in triplicate, and the results are presented as mean value ± standard deviation (SD). One-way ANOVA with Tukey’s HSD post hoc test was applied to evaluate whether the process factors had a significant effect (*p* < 0.05) on dependent variables in terms of *TPC*, *TAC*, *AC*, and phenolic compound contents. The values of dependent variables obtained at different levels of extraction factors were processed using principal component analysis (PCA) [34,35,36]. The Pearson correlation coefficient (*r*) was used to evaluate the strength of linear correlations between dependent variables. Response surface regression models were used to quantify the effects of process factors on *TPC* and *TAC*. Statistical analysis and process factor optimization were conducted using XLSTAT version 2019.1 (Addinsoft, New York, NY, USA) and STATISTICA version 10.0 (StatSoft Inc., Tulsa, OK, USA).

## 3. Results and Discussion

### 3.1. Total Phenolic Content (TPC)

First, the effect of the volume percentage of ethanol in the extraction solvent (*c_et_* = 50–100%) on *TPC* was evaluated. The experiments were performed at 30 °C for 30 min. Lower concentrations of ethanol were not used because a higher concentration of ethanol leads to improved long-term stability and therefore could be considered safer to add in food products. The results presented in Figure 1 highlight a decrease in *TPC* with an increase in *c_et_* and a significant effect (*p* < 0.05) of *c_et_*. High mean values of *TPC* were obtained for the hydroalcoholic extracts (21.8–32.2 mg GAE/g DM), and a significantly lower mean value was obtained in the case of the alcoholic extract (4.7 mg GAE/g DM). Accordingly, the optimal value of *c_et_* was considered to be 50%. 

Secondly, the effect of extraction time (*τ*) on *TPC* was assessed, keeping constant ethanol concentration (*c_et_* = 50%) and process temperature (*t* = 30 °C). The time range (*τ* = 5–180 min) was selected taking into account that a short period may not be enough for the depletion of the raw material into compounds of interest, while a high period could lead to compound degradation. The data shown in Figure 2 indicate that the mean values of *TPC* obtained at *τ* = 30–180 min (32.21–32.81 mg GAE/g DM) were similar (*p* > 0.05) and significantly higher (*p* < 0.05) than those corresponding to short extraction times (5 and 15 min), i.e., 21.38 and 22.23 mg GAE/g DM, respectively. The optimal value of *τ* was considered to be 30 min. 

Finally, the effect of extraction temperature (*t*) was studied at *c_et_* = 50% and *τ* = 30 min. As shown in Figure 3, an increase in *t* from 30 °C to 60 °C led to an increase in the mean value of *TPC*. The mean value of *TPC* obtained at *t* = 60 °C (37.23 mg GAE/g DM) was significantly higher (*p* < 0.05) than those obtained at lower temperatures (32.06–34.00 mg GAE/g DM). Moreover, the extract prepared using reflux heating (82.5 °C) had the lowest mean value of *TPC* (29.84 mg GAE/g DM). High temperatures are usually associated with phytocompound degradation [37]. Consequently, the optimal value of *t* was considered to be 60 °C. 

Accordingly, the maximum mean value of *TPC* (37.23 mg GAE/g DM) was obtained at the following optimal levels of process factors: *c_et_* = 50%, *τ* = 30 min, and *t* = 60 °C. The values of *TPC* obtained in this study, i.e., 4.50–37.40 mg GAE/g DM (1.39–11.59 mg GAE/g fresh fruit), were consistent with those reported in the related literature. Opriș et al. [38] applied ultrasound-assisted extraction (UAE) and reflux extraction (RE) to obtain blackthorn fruit extracts rich in polyphenols. For UAE, the maximum mean value of *TPC* (2.52 mg GAE/g DM) was obtained at an ethanol concentration of 40%, an extraction temperature of 67 °C, and an extraction time of 10 min. For RE, the maximum mean value of *TPC* (4.01 mg GAE/g DM) was obtained at an ethanol concentration of 30% and an extraction time of 45 min. Tahirovic et al. [39] studied the influence of methanol concentration (50% and 80%) and ethanol concentration (50% and 80%) on *TPC* using UAE. The mean values of *TPC* obtained at alcohol concentrations of 50% were higher than those found at alcohol concentrations of 80%, and the maximum mean value of *TPC* (30.20 mg GAE/g DM) was obtained at an ethanol concentration of 50%. Sikora et al. [2] reported a *TPC* mean value of 33.09 mg GAE/g DM (5.992 mg GAE/g fresh fruit) for the methanolic extract of blackthorn fruits. Veličković et al. [3] found a higher mean value of *TPC* for 50% ethanol (20.94 mg GAE/g fresh fruit) than those obtained for 50% methanol (17.69 mg GAE/g fresh fruit), 100% ethanol (15.33 mg GAE/g fresh fruit), 100% methanol (15.33 mg GAE/g fresh fruit), and water (12.17 mg GAE/g fresh fruit).

The mean values of *TPC* in the extract samples P1, P2,…, P13, i.e., *TPC_m,i_* (*i* = 1, 2,…, *N*, where *N* = 13), which are presented in Figure 1, Figure 2 and Figure 3, and their corresponding levels of process factors are summarized in Table 1. The effects of process factors (*c_et_*, *τ*, and *t*) on total phenolic content were quantified using the response surface regression model described by Equation (2), where the regression coefficients were obtained based on the experimental data specified in Table 1. The predicted values of total phenolic content (*TPC_pred,i_*) and related residuals (Δ*TPC_i_* = *TPC_m,I_* − *TPC_pred,i_*) are also presented in Table 1. The values of characteristic statistics of the regression model, i.e., *RMSE* defined by Equation (3), multiple *R*, multiple *R*^2^, adjusted *R*^2^, *F*, and *p*, which are also included in Table 1, indicate a good agreement between the experimental and predicted data.

The desirability function approach was applied to determine the optimal factor levels to maximize the response *Y* = *TPC_pred_*. A desirability function, *d*(*Y*), is defined by Equation (4) [40], where *L_Y_* = 4.71 mg GAE/g DM and *U_Y_* = 37.23 mg GAE/g DM are the lower and upper limits of process response. The predicted values of *Y* and *d*(*Y*) at different levels of extraction process factors, which are shown in Figure 4, indicate that the optimal levels of process factors to maximize the response *Y* are those obtained in the experimental study, i.e., *c_et_* = 50%, *τ* = 30 min, and *t* = 60 °C. Under these optimal conditions, the process response is *Y_opt_* = *TPC_pred,opt_* = 35.92 mg GAE/g DM and the desirability function is *d*(*Y_opt_*) = 0.96. Surface and contour plots of desirability, *d*(*Y = TPC_pred_*), depending on extraction process factors (*c_et_*, *τ*, and *t*) are shown in Figure 5.
(2)$$TPC_{pred}=-24.18+0.782c_{et}-0.008c_{et}{}^{2}+0.237\tau -0.001\tau^{2}+1.208t-0.010t^{2}$$
(3)$$RMSE=\sqrt{\frac{\sum_{i=1}^{N}\Delta TPC_{i}{}^{2}}{N}}=\sqrt{\frac{\sum_{i=1}^{13}{\left(TPC_{m,i}-TPC_{pred,i}\right)}^{2}}{13}}$$
(4)$$d\left(Y\right)=\left\{\begin{array}{ll} 0 & \;\mathrm{if}Y<L_{Y}\\ \frac{Y-L_{Y}}{U_{Y}-L_{Y}} & \;\mathrm{if}L_{Y}\leq Y\leq U_{Y}\\ 1 & \;\mathrm{if}Y>U_{Y} \end{array}\right.$$

### 3.2. Total Anthocyanin Content (TAC)

The effect of ethanol concentration in the extraction solvent (*c_et_* = 50–100%) on *TAC* at *t* = 30 °C and *τ* = 30 min (Figure 6) was similar to that on *TPC*. *TAC* and *TPC* were very strongly correlated (*r* = 0.993). As shown in Figure 6, the highest mean value of *TAC* (0.297 mg C3GE/g DM) was achieved for *c_et_* = 50%, and the lowest mean value (0.034 mg C3GE/g DM) was achieved for *c_et_* = 100%. 

The effect of extraction time (*τ* = 5–180 min) on *TAC* at *c_et_* = 50% and *t* = 30 °C is shown in Figure 7. The mean values of *TAC* increased significantly (from 0.214 to 0.333 mg C3GE/g DM) with increasing *τ* from 5 to 60 min, whereas the mean values of *TAC* for *τ* = 60–180 min (0.329–0.347 mg C3GE/g DM) were similar (*p* > 0.05). *TAC* and *TPC* were very strongly correlated (*r* = 0.921). Contrary to the results obtained for *TPC*, the anthocyanins require a longer time for a more efficient extraction. 

The effect of extraction temperature (*t* = 30–82.5 °C) on *TAC* at *c_et_* = 50% and *τ* = 30 min is shown in Figure 8. The mean values of *TAC* increased significantly (from 0.297 to 0.407 mg C3GE/g DM) with increasing *t* from 30 °C to 50 °C, were similar for *t* = 50 °C (0.407 mg C3GE/g DM) and *t* = 60 °C (0.415 mg C3GE/g DM), and decreased significantly (from 0.415 to 0.375 mg C3GE/g DM) with increasing *t* from 60 to 82.5 °C. 

The values of *TAC* obtained in this study, i.e., 0.034–0.415 mg C3GE/g DM (0.011–0.129 mg C3GE/g fresh fruit), were consistent with those reported in the literature. The anthocyanin content obtained by Tahirovic et al. [39] by UAE with 50% ethanol (0.973 mg C3GE/g DM) highlights the strong influence of ultrasonication on the extraction of anthocyanins. A slight decrease in the anthocyanin content (0.866 mg C3GE/g DM) was found with an increase in ethanol concentration in the extraction solvent (80%) [39]. Stanković et al. [41] reported a value of 1.973 mg C3GE/g DM for an extract obtained by UAE with 70% ethanol (at room temperature for 30 min). Veličković et al. [3] obtained higher mean values of *TAC* for an extraction with 50% ethanol (0.238 mg C3GE/g fresh fruit) compared to an extraction with absolute ethanol (0.11 mg C3GE/g fresh fruit) and water (0.12 mg C3GE/g fresh fruit).

The mean values of *TAC* in the extract samples P1, P2,…, P13, i.e., *TAC_m,i_* (*i* = 1, 2,…, *N*, where *N* = 13), which are presented in Figure 6, Figure 7 and Figure 8, and related levels of process factors are specified in Table 2. The effects of process factors (*c_et_*, *τ*, and *t*) on total anthocyanin content were quantified using the response surface regression model described by Equation (5), where the regression coefficients were estimated based on the experimental data summarized in Table 2. The predicted values of total anthocyanin content (*TAC_pred,i_*), corresponding residuals (Δ*TAC_i_* = *TAC_m,i_* − *TAC_pred,i_*), and characteristic statistics of the regression model, i.e., *RMSE* defined by Equation (6), multiple *R*, multiple *R*^2^, adjusted *R*^2^, *F*, and *p*, are also included in Table 2. The tabulated results indicate a very good agreement between the experimental and predicted data. 

For *d*(*Y* = *TAC_pred_*) calculated with Equation (4), where *L_Y_* = 0.034 mg G3GE/g DM and *U_Y_* = 0.415 mg G3GE/g DM, surface and contour plots of desirability depending on extraction process factors are similar to those shown in Figure 5.
(5)$$TAC_{pred}=\left(-0.28+7.82c_{et}-0.08c_{et}{}^{2}+2.37\tau -0.01\tau^{2}+12.1t-0.10t^{2}\right)\times 10^{-3}$$
(6)$$RMSE=\sqrt{\frac{\sum_{i=1}^{N}\Delta TAC_{i}{}^{2}}{N}}=\sqrt{\frac{\sum_{i=1}^{13}{\left(TAC_{m,i}-TAC_{pred,i}\right)}^{2}}{13}}$$

### 3.3. Antioxidant Capacity (AC)

The effects of the volume percentage of ethanol in the extraction solvent (*c_et_* = 50–100%) and extraction temperature (*t* = 30 °C and *t* = 60 °C) on *AC* in the extract samples obtained at *τ* = 30 min are shown in Figure 9. The effect of *c_et_* was similar to those on *TPC* and *TAC*. Moreover, *AC* (19.49–68.04 µmol TE/g DM, 6.04–21.09 µmol TE/g fresh fruit) was very strongly correlated with *TPC* (*r* = 0.997) and *TAC* (*r* = 0.988). The highest mean value of *AC* at *t* = 30 °C (56.82 µmol TE/g DM) was achieved for *c_et_* = 50%, and the lowest mean value (20.30 µmol TE/g DM) was achieved for *c_et_* = 100%. For *c_et_* = 50%, the mean value of *AC* at *t* = 60 °C (67.36 µmol TE/g DM) was significantly higher (by 19%) than that obtained at *t* = 30 °C. 

Tahirović et al. [39] found that the antioxidant activity increased with the decrease in the concentration of ethanol in the solvent mixture. Using the DPPH method, they obtained 140.80 µmol TE/g DM for 50% ethanol and 115.12 µmol TE/g DM for 80% ethanol. Sikora et al. [2] obtained 43.6 µmol TE/g fresh fruit for a methanolic extract from *P. spinosa* fruits. In the related literature, there are no studies on the application of the CUPRAC method for determining the antioxidant activity of *P. spinosa* fruit extracts. 

### 3.4. Chemical Profile of Extracts

Using HPLC-PDA analysis of the prepared extracts, up to five polyphenols from the twenty-three standard substances were identified (Table 3), five in the extracts prepared at 30 °C and four in the one obtained at 60 °C. The mean values ± SD of contents of protocatechuic acid, caffeic acid, vanillic acid, rutin hydrate, and quercetin in the extract samples obtained at *τ* = 30 min, at different levels of *c_et_* (50–100%) and *t* (30 °C and 60 °C), are summarized in Table 3. HPLC-PDA chromatograms corresponding to the extract samples specified in Table 3 (P1, P2, P3, P4, and P12) are shown in Figure 10. 

The mean values of protocatechuic acid content (*PA*) in the extracts were in the range of 2.08–4.47 mg/100 g DM; the highest mean value was found in the extract prepared at 60 °C with 50% ethanol, whereas the lowest one was found for the extract obtained at 30 °C with 100% ethanol. The effects of *c_et_* and *t* on *PA* were similar to those on *TPC*, *TAC*, and *AC*. Moreover, *PA* was very strongly correlated with *TPC* (*r* = 0.965), *TAC* (*r* = 0.914), and *AC* (*r* = 0.961). 

The values of caffeic acid content (*CA*) in the extracts ranged from 1.05 to 3.36 mg/100 g DM (0.32–1.04 mg/100 g fresh fruit). The mean value of *CA* obtained at *c_et_* = 50% and *t* = 30 °C (3.34 mg/100 g DM) was significantly higher (up to 3 times) than those obtained under other experimental conditions. Lower *CA* values (0.34 ± 0.04 mg/100 g DM) were reported by Pozzo et al. [15] for aqueous extracts prepared by batch extraction at room temperature for 2 h. Cosmulescu et al. [42] found *CA* values of 0.44 ± 0.02 mg/100 g fresh fruit for extracts prepared in methanol (70%) by UAE at 25 °C for 60 min, probably due to a higher solubility in methanolic solutions, the presence of an ultrasound field, or a longer extraction time.

The values of vanillic acid content (*VA*) in the extracts, i.e., 1.36–2.77 mg/100 g DM (0.42–0.86 mg/100 g fresh fruit), were lower than those found in methanolic extracts by Cosmulescu et al. [42] (3.14 ± 0.14 mg/100 g fresh fruit). The mean value of *VA* obtained in this study at *c_et_* =100% and *t* = 30 °C (1.46 mg/100 g DM) was significantly lower (up to 1.8 times) than those obtained under other experimental conditions. 

The values of rutin hydrate content (*RH*) in the extracts were in the range of 1.87–3.77 mg/100 g DM (0.58–1.17 mg/100 g fresh fruit). They were lower than those reported by Cosmulescu et al. [42] (4.86 ± 0.31 mg/100 g fresh fruit). The mean values of *RH* obtained in this study at *c_et_* = 50–75% and *t* = 30 °C (3.42–3.56 mg/100 g DM) were significantly higher (up to 90%) than those obtained under other experimental conditions. 

The values of quercetin content (*Q*) in the extracts obtained at *t* = 30 °C varied from 0.25 to 1.03 mg/100 g DM (0.08–0.32 mg/100 g fresh fruit). For *c_et_* = 50–75%, the mean values of *Q* (0.26–1.02 mg/100 g DM) increased with increasing *c_et_*. For *t* = 60 °C, the degradation of quercetin occurred, and its concentration decreased below the detection limit of the apparatus. The maximum values of *Q* obtained in this study (1.02 ± 0.01 mg/100 g DM) were consistent with those reported by Pozzo et al. [15] for an aqueous extract (0.99 ± 0.01 mg/100 g DM). Cosmulescu et al. [42] found significantly higher amounts of quercetin in methanolic extracts (4.86 ± 0.31 mg/100 g fresh fruit).

An FT-ICR-MS method was applied for the additional identification of the phenolic compounds quantified by HPLC-PDA. This method involved the direct infusion of the extract sample (without chromatographic separation) and was used only for qualitative analysis. FT-ICR spectrometers are the most advanced mass analyzers with resolving power with sub-ppm mass accuracy [43,44]. The SolariX software (https://www.bruker.com/en/products-and-solutions/mass-spectrometry/mrms/solarix.html, accessed on 24 September 2023) provides high-resolution predicted isotopic patterns, which can be compared to experimental spectral data. The predicted and measured values of the mass-to-charge ratio (*m*/*z*) for some phenolic compounds from extract samples P1 and P2 are summarized in Table 4, and related FT-ICR-MS spectra with negative ion mode ESI are shown in Appendix A. Phenolic compounds were successfully identified by comparing the predicted and measured values of *m*/*z* (percent errors from −0.00007% to 0.00006%). The results presented in Table 4 and Appendix A are consistent with those reported in the related literature. Pinacho et al. [1] used HPLC-MS with negative ion mode ESI and identified twenty-six phenolic compounds, including protocatechuic acid (*m*/*z* = 153), caffeic acid (*m*/*z* = 179), and quercetin (*m*/*z* = 301), in blackthorn extracts. Applying an HPLC-PDA-ESI-MS^3^ method, Magiera et al. [13] identified fifty-seven compounds in blackthorn fruit extracts, including protocatechuic acid, vanillic acid, rutin, and quercetin.

### 3.5. Multivariate Data Analysis

PCA was applied to evaluate the effects of the extraction conditions (*c_et_* and *t*) on *TPC*, *TAC*, *AC*, *PA*, *CA*, *VA*, *RH*, and *Q* in the extract samples obtained at *τ* = 30 min. The *PCA* results highlighted that only the eigenvalues corresponding to PC1 (5.84) and PC2 (1.89) were >1, and PC1 and PC2 explained 96.6% (73.0% + 23.6%) of the total variance. 

The results shown in Table 3, Figure 11 (PCA bi-plot), Table 5 (factor loadings), and Table 6 (correlation matrix) suggest the following:Depending on significant levels of factor loadings, the most important variables are *TPC*, *TAC*, *AC*, *PA*, *CA*, and *VA* for PC1 as well as *RH* and *Q* for PC2;Extract samples P1 (*c_et_* = 50% and *t* = 30 °C) and P12 (*c_et_* = 50% and *t* = 60 °C) had higher values of *TPC*, *TAC*, *AC*, *PA*, *CA*, and *VA* than sample P4 (*c_et_* = 100% and *t* = 30 °C) (discrimination on PC1 highlighted in Figure 11 using blue ellipses);Extract samples P2 (*c_et_* = 66.67% and *t* = 30 °C) and P3 (*c_et_* = 75% and *t* = 30 °C) had higher values of *RH* and *Q* than samples P4 (*c_et_* = 100% and *t* = 30 °C) and P12 (*c_et_* = 50% and *t* = 60 °C) (discrimination on PC2);*TPC*, *TAC*, *AC*, *PA*, *CA*, and *VA* were strongly directly correlated (0.722 ≤ *r* ≤ 0.995); *TPC*, *TAC*, and *AC* were inversely correlated with *Q* (−0.748 ≤ *r* ≤ −0.634); *RH* was directly correlated with *CA* (*r* = 0.642) and *Q* (*r* = 0.515).

The extract sample P12 obtained at *c_et_* = 50% and *t* = 60 °C (*τ* = 30 min) had the highest values of *TPC* (37.23 ± 0.21 mg GAE/g DM), *TAC* (0.42 ± 0.00 mg C3GE/g DM), *AC* (67.36 ± 0.62 μmole TE/g DM), *PA* (4.47 ± 0.06 mg/100 g DM), and *VA* (2.65 ± 0.11 mg/100 g DM), but lower values of *RH* (1.87 ± 0.00 mg/100 g DM) and *Q* (0.00 ± 0.00 mg/100 g DM) than those obtained under other experimental conditions. Accordingly, at higher levels of process temperature, the extraction of protocatechuic acid and vanillic acid was enhanced, but the flavonoids (rutin hydrate and quercetin) were degraded. The extract sample P3 obtained at *c_et_* = 75% and *t* = 30 °C (*τ* = 30 min) had the highest values of flavonoid content (*RH* = 3.56 ± 0.02 mg/100 g DM and *Q* = 1.02 ± 0.01 mg/100 g DM).

## 4. Conclusions

This study highlighted that the optimal extraction of different types of phytocompounds requires special experimental conditions depending on their chemical and thermal stability as well as molecular mass. 

All tested parameters, i.e., ethanol concentration in the extraction solvent (*c_et_*), process temperature (*t*), and operating time (*τ*), affected the performance of the extraction process. A synergistic effect of obtaining high total contents of polyphenols (*TPC*) and anthocyanins (*TAC*), high contents of protocatechuic acid (*PA*) and vanillic acid (*VA*), and a good antioxidant capacity (*AC*) was achieved for the extraction with 50% ethanol at 60 °C for 30 min. At a higher level of process temperature, the extraction of protocatechuic acid and vanillic acid was enhanced, but the flavonoids (rutin hydrate and quercetin) were degraded. If a high amount of flavonoids is desired, a lower temperature should be used. *TPC*, *TAC*, *AC*, and phenolic acid contents (*PA*, *CA*, and *VA*) in the extract samples obtained at *c_et_* = 50–100%, *t* = 30–60 °C, and *τ* = 30 min were strongly directly correlated. Flavonoid contents (*RH* and *Q*) as well as *RH* and *CA* were directly correlated, whereas *TPC*, *TAC*, and *AC* were inversely correlated with *Q*.

## Figures and Tables

**Figure 1 antioxidants-12-01897-f001:**
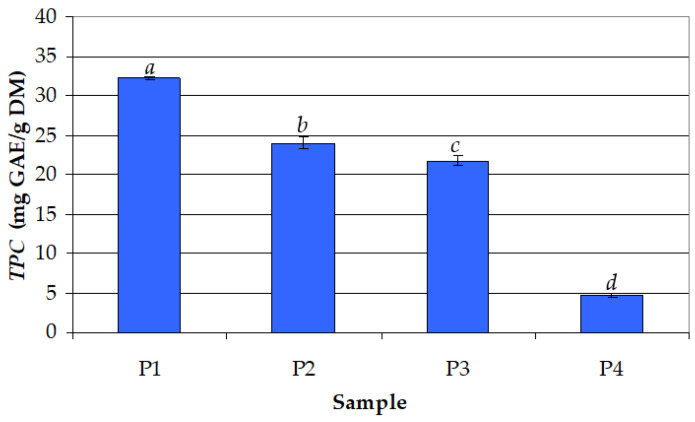
Effect of ethanol concentration in the extraction solvent (*c_et_*) on the total phenolic content (*TPC*) for different samples (process temperature: *t* = 30 °C; extraction time: *τ* = 30 min): (P1) *c_et_* = 50%; (P2) *c_et_* = 66.67%; (P3) *c_et_* = 75%; (P4) *c_et_* = 100%; different letters indicate a significant difference (*p* < 0.05).

**Figure 2 antioxidants-12-01897-f002:**
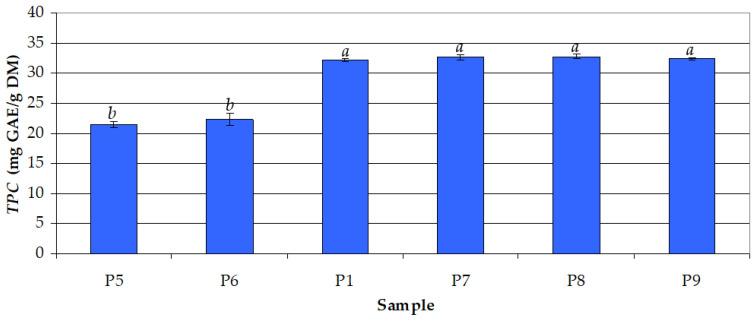
Effect of extraction time (*τ*) on the total phenolic content (*TPC*) for different samples (ethanol concentration: *c_et_* = 50%; process temperature: *t* = 30 °C): (P5) *τ* = 5 min; (P6) *τ* = 15 min; (P1) *τ* = 30 min; (P7) *τ* = 60 min; (P8) *τ* = 120 min; (P9) *τ* = 180 min; different letters indicate a significant difference (*p* < 0.05).

**Figure 3 antioxidants-12-01897-f003:**
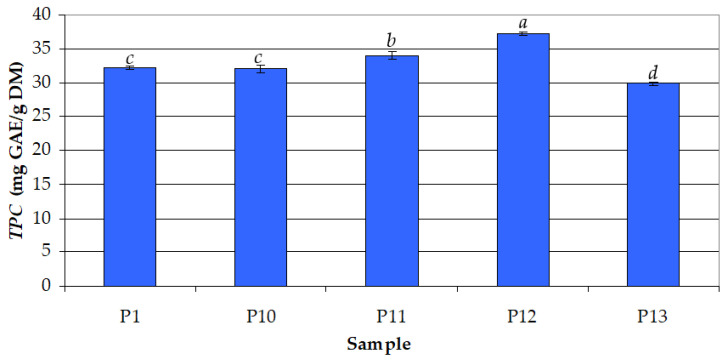
Effect of process temperature (*t*) on the total phenolic content (*TPC*) for different samples (ethanol concentration: *c_et_* = 50%; extraction time: *τ* = 30 min): (P1) *t* = 30 °C; (P10) *t* = 40 °C; (P11) *t* = 50 °C; (P12) *t* = 60 °C; (P13) *t* = 82.5 °C; different letters indicate a significant difference (*p* < 0.05).

**Figure 4 antioxidants-12-01897-f004:**
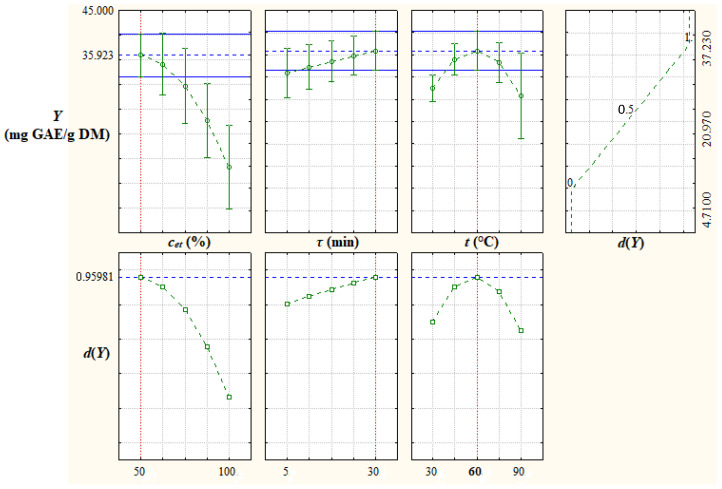
Predicted values of total phenolic content, *Y* = *TPC_pred_*, and desirability, *d*(*Y*), at different levels of extraction process factors; *c_et_*, volume percentage of ethanol in the extraction solvent; *τ*, extraction time; *t*, extraction temperature.

**Figure 5 antioxidants-12-01897-f005:**
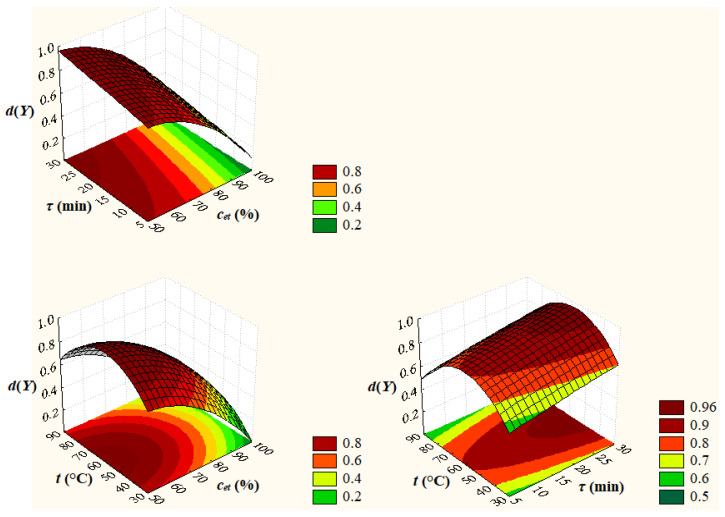
Surface and contour plots of desirability, *d*(*Y* = *TPC_pred_*), depending on extraction process factors; *c_et_*, volume percentage of ethanol in the extraction solvent; *τ*, extraction time; *t*, extraction temperature.

**Figure 6 antioxidants-12-01897-f006:**
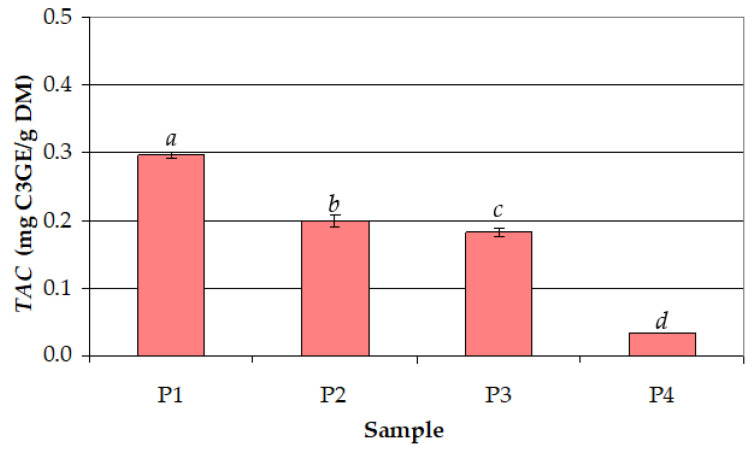
Effect of ethanol concentration (*c_et_*) in the extraction solvent on the total anthocyanin content (*TAC*) for different samples (process temperature: *t* = 30 °C; extraction time: *τ* = 30 min): (P1) *c_et_* = 50%; (P2) *c_et_* = 66.67%; (P3) *c_et_* = 75%; (P4) *c_et_* = 100%; different letters indicate a significant difference (*p* < 0.05).

**Figure 7 antioxidants-12-01897-f007:**
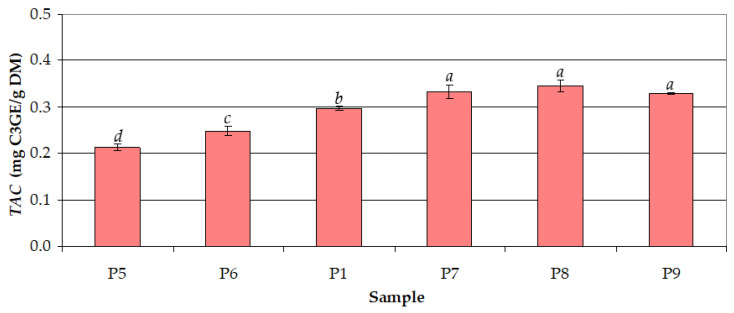
Effect of extraction time (*τ*) on the total anthocyanin content (*TAC*) for different samples (ethanol concentration: *c_et_* = 50%; process temperature: *t* = 30 °C): (P5) *τ* = 5 min; (P6) *τ* = 15 min; (P1) *τ* = 30 min; (P7) *τ* = 60 min; (P8) *τ* = 120 min; (P9) *τ* = 180 min; different letters indicate a significant difference (*p* < 0.05).

**Figure 8 antioxidants-12-01897-f008:**
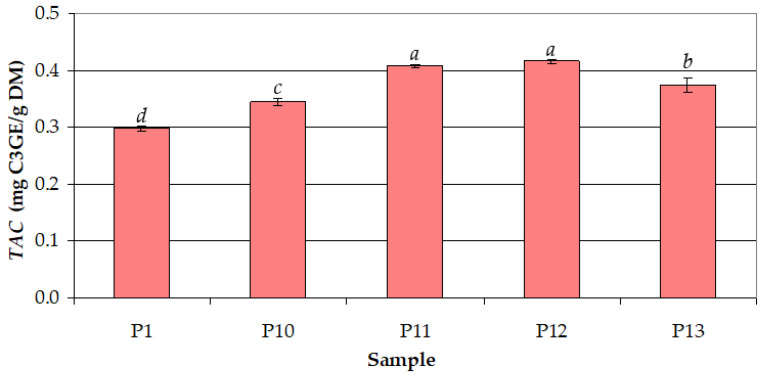
Effect of process temperature (*t*) on the total anthocyanin content (*TAC*) for different samples (ethanol concentration: *c_et_* = 50%; extraction time: *τ* = 30 min): (P1) *t* = 30 °C; (P10) *t* = 40 °C; (P11) *t* = 50 °C; (P12) *t* = 60 °C; (P13) *t* = 82.5 °C; different letters indicate a significant difference (*p* < 0.05).

**Figure 9 antioxidants-12-01897-f009:**
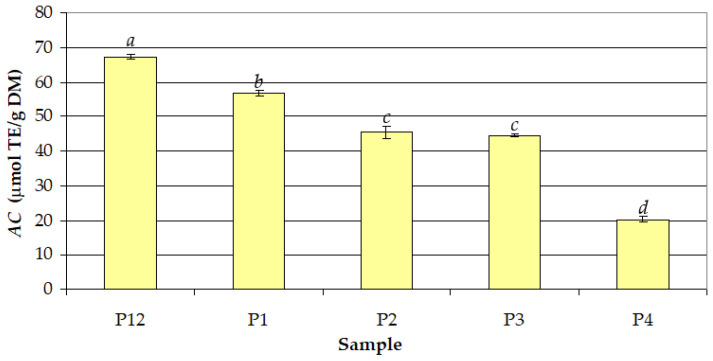
Effects of volume percentage of ethanol in the extraction solvent (*c_et_*) and process temperature (*t*) on the antioxidant capacity (*AC*) of different samples (extraction time: *τ* = 30 min): (P12) *c_et_* = 50% and *t* = 60 °C; (P1) *c_et_* = 50% and *t* = 30 °C; (P2) *c_et_* = 66.67% and *t* = 30 °C; (P3) *c_et_* = 75% and *t* = 30 °C; (P4) *c_et_* = 100% and *t* = 30 °C; different letters indicate a significant difference (*p* < 0.05).

**Figure 10 antioxidants-12-01897-f010:**
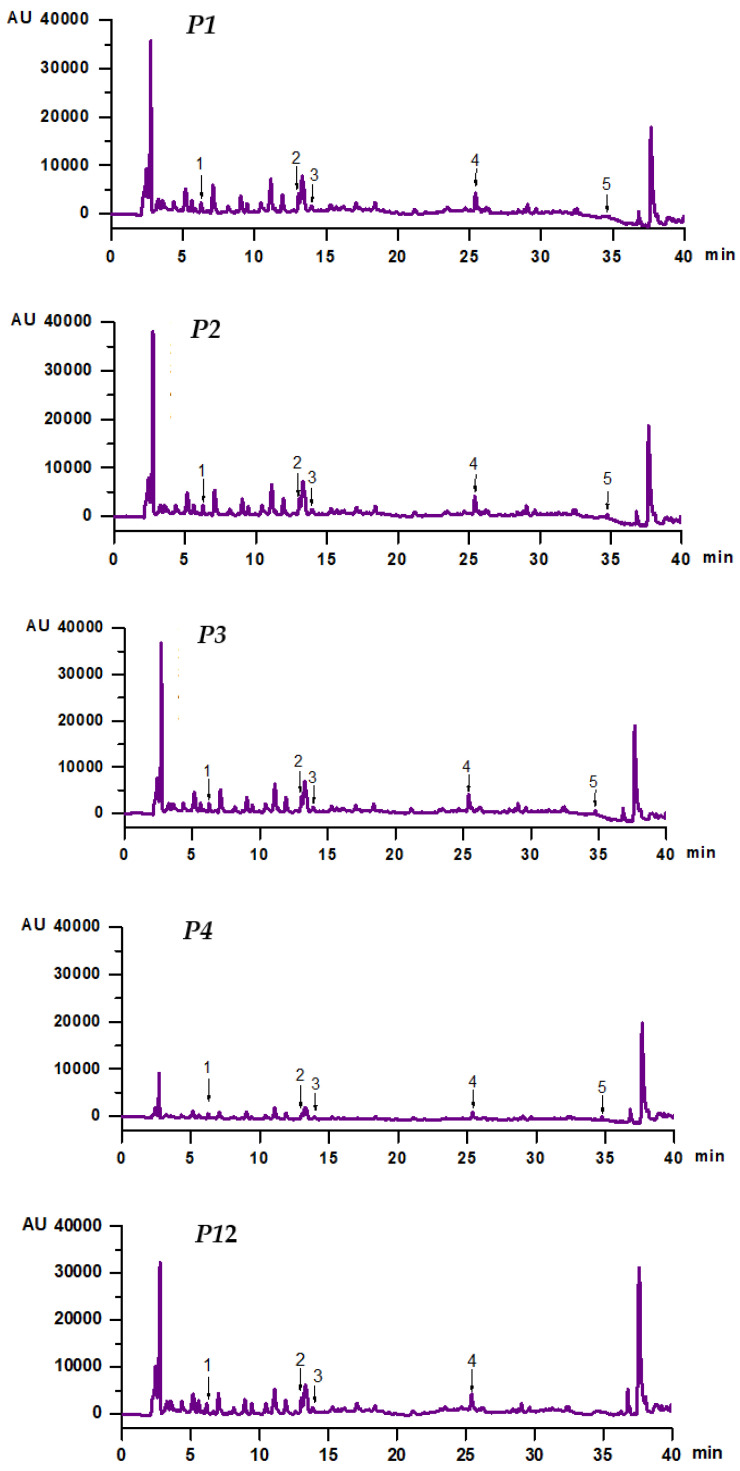
HPLC-PDA chromatograms (absorbance *vs.* retention time) corresponding to extract samples P1 (*c_et_* = 50% and *t* = 30 °C), P2 (*c_et_* = 66.67% and *t* = 30 °C), P3 (*c_et_* = 75% and *t* = 30 °C), P4 (*c_et_* = 100% and *t* = 30 °C), and P12 (*c_et_* = 50% and *t* = 60 °C) obtained at *τ* = 30 min: (1) protocatechuic acid; (2) caffeic acid; (3) vanillic acid; (4) rutin hydrate; (5) quercetin; *c_et_*, volume percentage of ethanol in the extraction solvent; *t*, extraction temperature; *τ*, extraction time.

**Figure 11 antioxidants-12-01897-f011:**
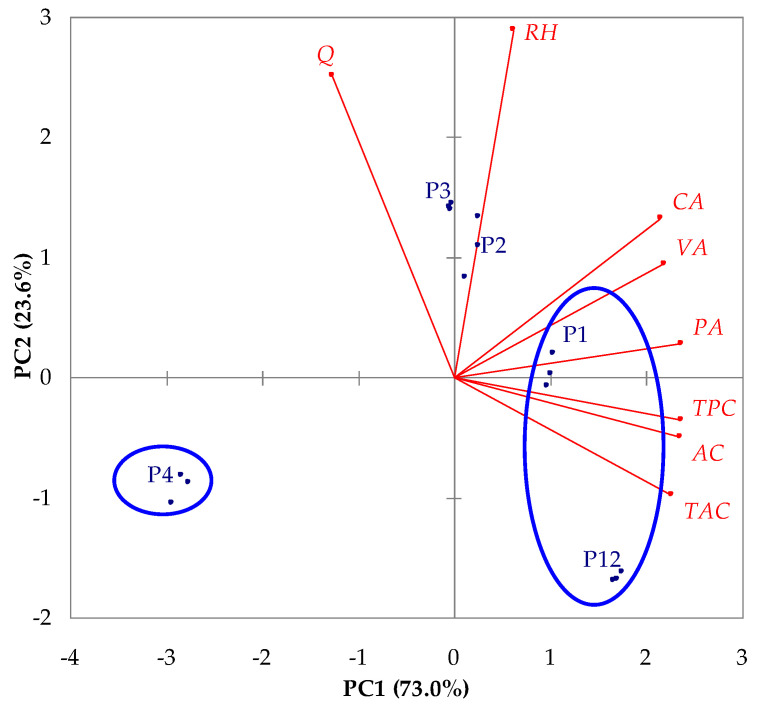
Projections of variables (*TPC*, *TAC*, *AC*, *PA*, *CA*, *VA*, *RH*, and *Q*) and samples (P1, P2, P3, P4, and P12) on the factor-plane PC1–PC2; *TPC*, total phenolic content; *TAC*, total anthocyanin content; *AC*, antioxidant capacity; *PA*, protocatechuic acid content; *CA*, caffeic acid content; *VA*, vanillic acid content; *RH*, rutin hydrate content; *Q*, quercetin content; the levels of process factors for extract samples are specified in Table 3.

**Table 1 antioxidants-12-01897-t001:** Experimental and predicted values of total phenolic content at different levels of extraction process factors and relevant statistics of regression model.

*i*	Sample	*c_et_* (%)	*τ* (min)	*t* (°C)	*TPC_m,i_* (mg GAE/g DM)	*TPC_pred,i_* (mg GAE/g DM)	Δ*TPC_i_* (mg GAE/g DM)
1	P1	50	30	30	32.21	27.49	4.7
2	P2	66.67	30	30	23.96	24.49	−0.5
3	P3	75	30	30	21.75	21.28	0.5
4	P4	100	30	30	4.71	4.77	−0.1
5	P5	50	5	30	21.38	22.44	−1.1
6	P6	50	15	30	22.23	24.61	−2.4
7	P7	50	60	30	32.69	31.89	0.8
8	P8	50	120	30	32.81	35.30	−2.5
9	P9	50	180	30	32.44	31.52	0.9
10	P10	50	30	40	32.06	32.36	−0.3
11	P11	50	30	50	34.00	35.17	−1.2
12	P12	50	30	60	37.23	35.92	1.3
13	P13	50	30	82.5	29.84	30.09	−0.2
						*RMSE*	1.767
						Multiple *R*	0.977
						Multiple *R*^2^	0.955
						Adjusted *R*^2^	0.909
						*F*	21.06
						*p*	0.001

*c_et_*, volume percentage of ethanol in the extraction solvent; *τ*, extraction time; *t*, extraction temperature; *TPC_m,i_*, mean experimental values of total phenolic content; *TPC_pred,i_*, predicted values of total phenolic content calculated using Equation (2); Δ*TPC_i_* = *TPC_m,i_* − *TPC_pred,i_*, residual; *RMSE*, root mean square error defined by Equation (3); multiple *R*, coefficient of multiple correlation; multiple *R*^2^, coefficient of multiple determination; adjusted *R*^2^, adjusted coefficient of multiple determination, *F*, *F*-value; *p*, *p*-value.

**Table 2 antioxidants-12-01897-t002:** Experimental and predicted values of total anthocyanin content at different levels of extraction process factors and relevant statistics of regression model.

*i*	Sample	*c_et_* (%)	*τ* (min)	*t* (°C)	*TAC_m,i_* (mg C3GE/g DM)	*TAC_pred,i_* (mg C3GE/g DM)	Δ*TAC_i_* (mg C3GE/g DM)
1	P1	50	30	30	0.297	0.279	0.018
2	P2	66.67	30	30	0.199	0.211	−0.012
3	P3	75	30	30	0.182	0.172	0.010
4	P4	100	30	30	0.034	0.035	−0.001
5	P5	50	5	30	0.214	0.226	−0.012
6	P6	50	15	30	0.249	0.249	0.000
7	P7	50	60	30	0.333	0.325	0.007
8	P8	50	120	30	0.347	0.362	−0.015
9	P9	50	180	30	0.329	0.323	0.005
10	P10	50	30	40	0.345	0.351	−0.006
11	P11	50	30	50	0.407	0.398	0.008
12	P12	50	30	60	0.415	0.420	−0.004
13	P13	50	30	82.5	0.375	0.374	0.000
						*RMSE*	0.009
						Multiple *R*	0.996
						Multiple *R*^2^	0.992
						Adjusted *R*^2^	0.983
						*F*	118.5
						*p*	0.00001

*c_et_*, volume percentage of ethanol in the extraction solvent; *τ*, extraction time; *t*, extraction temperature; *TAC_m,i_*, mean experimental values of total anthocyanin content; *TAC_pred,i_*, predicted values of total anthocyanin content calculated using Equation (5); Δ*TAC_i_* = *TAC_m,i_* − *TAC_pred,i_*, residual; *RMSE*, root mean square error defined by Equation (6); multiple *R*, coefficient of multiple correlation; multiple *R*^2^, coefficient of multiple determination; adjusted *R*^2^, adjusted coefficient of multiple determination, *F*, *F*-value; *p*, *p*-value.

**Table 3 antioxidants-12-01897-t003:** HPLC-PDA polyphenolic profile of blackthorn extracts obtained at an extraction time of 30 min.

Sample	*c_et_* (%)	*t* (°C)	*PA* (mg/100 g DM)	*CA* (mg/100 g DM)	*VA* (mg/100 g DM)	*RH* (mg/100 g DM)	*Q* (mg/100 g DM)
P12	50	60	4.47 ± 0.06 a	2.96 ± 0.02 d	2.65 ± 0.11 a	1.87 ± 0.00 b	-
P1	50	30	3.97 ± 0.02 b	3.34 ± 0.02 a	2.37 ± 0.03 b	3.42 ± 0.17 a	0.26 ± 0.00 d
P2	66.67	30	3.83 ± 0.02 c	3.15 ± 0.03 b	2.55 ± 0.02 a	3.43 ± 0.33 a	0.83 ± 0.01 b
P3	75	30	3.75 ± 0.03 d	3.04 ± 0.02 c	2.51±0.01 ab	3.56 ± 0.02 a	1.02 ± 0.01 a
P4	100	30	2.08 ± 0.03 e	1.06 ± 0.01 e	1.46 ± 0.10 c	1.96 ± 0.09 b	0.68 ± 0.01 c

*c_et_*, volume percentage of ethanol in the extraction solvent; *t*, extraction temperature; *PA*, protocatechuic acid content; *CA*, caffeic acid content; *VA*, vanillic acid content; *RH*, rutin hydrate content; *Q*, quercetin content; *DM*, dry matter content; different letters in the same column indicate a significant difference (*p* < 0.05).

**Table 4 antioxidants-12-01897-t004:** Predicted and measured values of mass-to-charge ratio for some phenolic compounds from blackthorn extract samples identified by FT-ICR-MS analysis with negative ion mode ESI.

Compound	Molecular Formula	Mass-to-Charge Ratio (*m/z*)		
		Predicted	Measured in Sample P1	Measured in Sample P2
Protocatechuic acid (PA)	C_7_H_6_O_4_	153.019332	153.019354	153.019339
Vanillic acid (VA)	C_8_H_8_O_4_	167.034982	167.035008	167.034891
Caffeic acid (CA)	C_9_H_8_O_4_	179.034982	179.034941	179.034941
Quercetin (Q)	C_15_H_10_O_7_	301.035376	301.035407	301.035380
Rutin (R)	C_27_H_30_O_16_	609.146108	609.146473	609.145659

**Table 5 antioxidants-12-01897-t005:** Factor loadings.

Variable	PC1	PC2
Total phenolic content (*TPC*)	**0.992**	−0.110
Total anthocyanin content (*TAC*)	**0.950**	−0.308
Antioxidant capacity (*AC*)	**0.986**	−0.155
Protocatechuic acid content (*PA*)	**0.989**	0.090
Caffeic acid content (*CA*)	**0.900**	0.423
Vanillic acid content (*VA*)	**0.916**	0.301
Rutin hydrate content (*RH*)	0.257	**0.920**
Quercetin content (*Q*)	−0.535	**0.798**

PC, principal component; significant values of factor loadings are highlighted in bold.

**Table 6 antioxidants-12-01897-t006:** Correlation matrix.

Variable	*TPC*	*TAC*	*AC*	*PA*	*CA*	*VA*	*RH*	*Q*
** *TPC* **	**1**	**0.975**	**0.995**	**0.965**	**0.851**	**0.860**	0.170	**−0.634**
** *TAC* **	**0.975**	**1**	**0.985**	**0.914**	**0.722**	**0.780**	−0.042	**−0.748**
** *AC* **	**0.995**	**0.985**	**1**	**0.961**	**0.820**	**0.856**	0.112	**−0.650**
** *PA* **	**0.965**	**0.914**	**0.961**	**1**	**0.919**	**0.957**	0.305	−0.427
** *CA* **	**0.851**	**0.722**	**0.820**	**0.919**	**1**	**0.929**	**0.642**	−0.165
** *VA* **	**0.860**	**0.780**	**0.856**	**0.957**	**0.929**	**1**	0.440	−0.186
** *RH* **	0.170	−0.042	0.112	0.305	**0.642**	0.440	**1**	**0.515**
** *Q* **	**−0.634**	**−0.748**	**−0.650**	−0.427	−0.165	−0.186	**0.515**	**1**

*TPC*, total phenolic content; *TAC*, total phenolic content; *AC*, antioxidant capacity; *PA*, protocatechuic acid content; *CA*, caffeic acid content; *VA*, vanillic acid content; *RH*, rutin hydrate content; *Q*, quercetin content; significant values of correlation coefficients at a significance level *α* = 0.05 (two-tailed test) are highlighted in bold.

## Data Availability

No new data were created or analyzed in this study. Data sharing is not applicable to this article.

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
