# Peer review of "Effects of Extraction Process Factors on the Composition and Antioxidant Activity of Blackthorn (Prunus spinosa L.) Fruit Extracts"

_antioxidants, 2023, doi:10.3390/antiox12101897_

Round 1

Reviewer 1 Report

The authors conducted studies on Prunus spinosa L. fruits to determine conditions for the extraction of phenolic compounds. The research topic is interesting from a scientific point of view. The stone fruits of this plant species are used for food and are highly valued in the folk medicine of many countries. The material of the article is arranged sequentially, the results are presented clearly. However, I have some remarks for the authors.

In my opinion, the title does not correspond to either the content or the aim (p. 3, lines 106-111) of the research. The aim of the conducted research was not to determine the amount of phenolic compounds in stone fruits but to specify phytochemical research methods by determining the optimal temperature, extraction time, and ethanol concentration. So, the title must reflect the content.

Abstract. The results of the work are briefly indicated in the abstract. However, the incorrect term 'were denatured' is used, throughout the article as well.

Keywords are appropriate, although authors should also include terms more relevant to research methods optimization. This will help readers to find the publication faster.

Introduction. This chapter is quite detailed, as many as 27 literature sources are analysed and the problem is discussed.

Materials and methods. This chapter needs to be corrected. The Plant material section must be supplemented. At what temperature were the fruits dried, how much weight was lost after drying, were the stones removed or not etc.

Results and discussion. The research results are presented consistently and relevant works of other authors are cited. The applied statistical analysis methods confirm the reliability of the obtained results. In my opinion, the term denaturation is used incorrectly throughout the text. Denaturation (de... + natura – innate properties, nature) changes the natural, biologically active conformation of biopolymer molecules. Proteins and nucleic acids can be denatured. Therefore, changes in the structure of flavonoids could be described by the terms degradation or fragmentation. On the other hand, was it determined what compounds were formed from the mentioned flavonoids?

Conclusions. This section briefly describes the results, but the terms - species extraction and denaturation - should be corrected.

Minor notes.

How to understand the phrases: shrub fruits (p. 1, line 41), species extraction (p.12, line 431-432)?

When the name of a genus or species of a plant is mentioned for the first time, the initials of the authors must be indicated, Prunus spinosa L. Further in the text, the name of the genus must be abbreviated to the first letter and the author's name should not be written, P. spinosa.

References. The article cites 39 sources, but I would recommend including and analysing the following publication in the discussion section:

 Biesaga M.  Influence of extraction methods on stability of flavonoids. Journal of Chromatography A, 1218 (2011) 2505–2512   doi:10.1016/j.chroma.2011.02.059

Reviewer 2 Report

The paper titled "Superior valorization of forgotten berries in Romania – Prunus spinosa L." assesses the impact of various extraction conditions on the production of polyphenolic compounds and their antioxidant activity. Following a comprehensive review, my feedback regarding the manuscript is generally positive. However, before granting acceptance for publication in the journal Antioxidants, several adjustments are necessary concerning the following suggestions:

1. It is noteworthy that the Response Surface Methodology (RSM) plays a pivotal role in optimizing the extraction conditions to achieve the most efficient isolation of polyphenols. In light of this, I kindly request that the Authors employ the RSM method for optimizing the polyphenol content by varying parameters such as ethanol concentration in water, extraction time, and extraction temperature. The article will attain substantive value by doing so, rendering the optimization process results more credible.

2. Considering various parameters of ethanol concentration in water and extraction temperature, the Authors also examined the content of dominant polyphenols using the HPLC method. However, I harbor reservations concerning the qualitative composition of the analyzed extracts. As depicted in Table 1, among the five identified polyphenols, three of the dominant compounds in the extract are categorized as phenolic acids (namely vanillic acid, protocatechuic acid, and caffeic acid). The Authors identified the components of the tested extracts by comparing their retention times and UV spectra with those obtained for standard substances. Nevertheless, it has become common practice to bolster such identification through additional analysis employing Mass Spectrometry (MS) [Magiera, A.; et al. Molecules 2022, 27, 1691. https://doi.org/10.3390/molecules27051691]. In light of this, I kindly request two modifications: Firstly, please append an HPLC chromatogram to the manuscript showcasing the separation of polyphenols within the tested extracts. Secondly, kindly present the outcomes of the LC-MS analysis conducted on the tested extracts. Including these additional identification elements will undoubtedly enhance the reliability of the results presented in the paper.

Round 2

Reviewer 2 Report

The article marked with ID 2657978 has been partially corrected, but I still have a few comments for the Authors:

·         Please include information in the experimental section regarding the LC-MS analysis of extracts P1 and P12. Additionally, in the Results and Discussion section, present a table with identified polyphenolic compounds from extracts P1 and P12. Presenting the characteristics of each compound in the form of UVmas values and MS fragmentation ensures the accuracy of identifying the components in the tested extracts. Please also conduct a discussion, comparing it with other literature data concerning the qualitative composition of polyphenols detected in the examined extracts. Kindly provide an argumentation for why phenolic acids such as protocatechuic, vanillic, and caffeic, and flavonoids such as rutin and quercetin were chosen for subsequent LC-PDA measurements. These compounds, as evidenced by the LC-PDA chromatograms, are not the dominant components in the tested extracts P1, P2, and P3.

·         Please explain why the Supplementary materials file was not attached to the article. Kindly complete this and include an LC-UV or MS chromatogram of one of the extracts in those materials. This will eliminate any doubts regarding the completion of the analysis.

·         Please supplement your results with a graphical representation of the plane in the Response Surface Methodology (RSM) model, including the obtained optimal extraction points.

Round 3

Reviewer 2 Report

The Authors have largely addressed the recommendations of the reviewer in revising the manuscript. Hence, it is now suitable for publication in its present form.